# Reduced visual context effects in global motion processing in depression

**Grace E. Murray**[1,2,3], **Daniel J. Norton** [1,2,3,4] *

**1** Department of Psychological and Brain Sciences, Boston University, Boston, MA, United States of America, **2** Department of Psychology, Williams College, Williamstown, MA, United States of America, **3** McLean Hospital, Belmont, MA, United States of America, **4** Department of Psychology, Gordon College, Wenham, MA, United States of America

* daniel.norton@gordon.edu

## Abstract

Research supports abnormal inhibitory visual motion processing in adults with remitted and current depression, but all studies to date have used paradigms with simple grating stimuli. Global motion processing, where multiple motion signals must be integrated, has not been explored in depression, nor have inhibitory processes within that domain. Depressed participants (n = 46) and healthy controls (n = 28) completed a direction discrimination task featuring a random dot pattern stimulus. Various signal (rightward or leftward dots) to noise (dots with randomly assigned directions) ratios modulated task difficulty. Metrics of global center surround suppression and facilitation were calculated. Accuracy in the baseline condition (i.e., no surrounding annulus) was not significantly different between depressed and healthy participants. Global center surround suppression and facilitation were not significantly different between healthy and depressed participants overall. When limiting the sample to unmedicated individuals, depressed participants (n = 27) showed a reduced global center surround suppression effect compared to controls, and there was no difference in global center surround facilitation. While global motion processing is intact in depression, abnormal center surround suppression effects in depression do extend to global motion stimuli. These alterations may be mitigated by the psychotropic medications taken by some subjects in our depressed sample. Future studies should explore the mechanisms underlying these effects.

## Introduction

Major depressive disorder (MDD) is among the most common psychiatric disorders [1], with substantial literature describing both its mental [2, 3] and biological [4, 5] characteristics. Studying visual processing in depressed individuals has become a promising avenue for bridging psychological and biological phenomena in this disorder due to the highly quantifiable and well-understood nature of the visual system [6–8]. Assessing how visual perception in depressed individuals differs from that of healthy controls can offer insight into functional differences of the visual system, contributing to broader efforts to identify markers of depression to aid in early diagnosis and intervention [8]. Further, some literature theorizes that visual

**Funding:** This work was supported by a grant from McLean Hospital and the Rossano family to DJN,

as well as internal funding from Gordon College and Williams College. This research was also supported in part by NINDS F31 NS07682 to DJN, and a grant from the Supporting Structures: Innovative Partnerships to Enhance Bench Science at CCCU Member Institutions program, run by Scholarship and Christianity in Oxford, the UK subsidiary of the Council for Christian Colleges and Universities, with funding by the John Templeton Foundation and the MJ Murdock Charitable Trust. There was no additional external funding received for this study. The funders had no role in study design, data collection and analysis, decision to publish, or preparation of the manuscript.

**Competing interests:** The authors have declared that no competing interests exist.

perception abnormalities contribute to depressive symptoms by reducing the likelihood that perceptual data will activate neural reward circuitry [6]. If this is the case, improving our understanding of these perceptual differences in depressed vs. healthy brains may be a first step toward improving clinical interventions.

One fruitful area of research has focused on visual forms of inhibitory processing in depression [9], with most studies examining these effects within visual motion perception [10–12]. These studies have been interpreted as being related to a phenomenon called center surround suppression (CSS). In neurons with center-surround receptive fields, stimulation of the center of the field increases the firing rate of the neuron, and stimulation of the region surrounding that in space (the surround) decreases the firing rate [13]. While CSS was discovered in single-cell recording studies [7, 14] at the level of an individual neuron, it also occurs at the neuronal population level, and its consequences can be observed in perceptual tasks [15, 16]. When tasked with discriminating the direction of motion of a drifting grating, participants' accuracy decreases as the diameter of the grating increases, suggesting that as the stimulus encroached on the surround portion of the visual field, perception was suppressed [16]. Some stimuli can also have a facilitatory effect on neuronal activation (center surround facilitation, CSF) [17]. Whether an interaction will be facilitatory or suppressive depends on various stimulus characteristics—for motion perception, facilitation tends to occur when the surrounding stimulus is in the opposite direction of the center, and suppression occurs when they move in the same direction [18]. The present study measured both suppressive and facilitatory effects in depressed and healthy adults.

## Abnormalities in center surround interaction in depression

Multiple studies that employed psychophysical paradigms to elicit perceptual evidence of CSS have shown abnormalities in individuals who were currently or previously depressed [10–12]. Using a drifting grating stimulus, Golomb et al. [11] showed that recovered-depressed individuals demonstrated enhanced motion perception of large, high contrast stimuli, suggesting reduced CSS in that group. A recent study [12] also investigated the presumed perceptual consequences of CSS in acutely depressed individuals using a drifting sinusoidal grating and found that the degree of suppression was negatively correlated with depression severity, and that suppression indices were reduced in the depressed sample (though this seems to have been driven by something other than CSS—see discussion of the present paper). That study also used magnetic resonance spectroscopy (MRS) and found decreased GABA levels in the middle temporal area (MT) of their depressed sample. Interestingly, GABA levels were correlated with visual motion performance in healthy, but not depressed, participants. Norton et al. [10] measured CSS in high and low contrast stimuli (horizontally drifting gratings) and found that currently depressed individuals had significantly more CSS than controls when viewing high-contrast stimuli, but when viewing low-contrast stimuli, this effect vanished and trended in the opposite direction (though these differences only emerged at certain presentation times). These three studies consistently support abnormal center surround interaction (CSI) effects in depression, although further research is needed to clarify the nature of these abnormalities due to inconsistencies across studies (differences in current diagnostic status, stimuli, etc.).

Prior basic and clinical research has distinguished between motion processing as being local or global in nature [19–21]. For example, the receptive field of particular motion processing cells in V1 or MT may have access to only the "local motion" of a part of an object but not the whole object itself, the "global motion" [18]. Or, in the case of a random dot pattern, there may be many local motions, many of which are random, while the direction of global coherent

motion is still apparent [19], forcing the perceiving organism to use global motion signals to perform a task. This distinction of global versus local motion has also been applied in the realm of brain disorders, where one study showed that direction discrimination was impaired in global, but not local motion in schizophrenia [20]. In the present study, we refer to global motion as that which computationally requires the integration of multiple local motion signals to correctly perform the task. Local motion is that which can be correctly perceived by relying on any local receptive field, as in a grating stimulus. Existing research regarding CSS in depression has been limited to studies using grating stimuli, local motion tasks [10–12], and whether inhibitory processing in depression is altered in a global motion processing task has not yet been established. Furthermore, existing studies have varied with regard to the medication status of their participants [11, 12]. It is theorized that GABA is related to CSS as the primary inhibitory neurotransmitter in the brain [22, 23], and GABA concentrations in the occipital cortex may be impacted by even non-GABAergic medications [1–3, 24–26].

In the present study, we applied a global motion task [27] that requires motion integration of conflicting signals [20, 28]. The task, discrimination of the direction of motion of coherently moving dots within an RDP, required motion discrimination of a field of moving dots, rather than a cohesive grating. To examine whether global motion processing was intact in MDD, we first examined performance on this RDP task as a function of motion coherence (the "Baseline" condition). We anticipated that this function would be intact in MDD, although it has been shown to be altered in other disorders such as schizophrenia in some, but not all, studies [20, 29]. Then, to examine whether inhibitory processing was affected in a global motion task, we added an annulus to the RDP stimulus that affected perception of the central field. Our primary outcomes here were the CSS and CSF effects [30]. Because there is no pre-existing literature on CSS and CSF in a global motion task in depression, and because of the conflicting results in prior studies using local motion stimuli, we did not have a clear hypothesis on the direction of abnormalities in CSS or CSF of global motion perception in depression.

### Aims of the study

The key aims of this study were to explore center surround interactions in global motion processing in depression, and to test the integrity of global motion processing itself in depression. We were additionally interested in the effects of psychotropic medications, and we examined differences in center surround interactions in medicated vs. unmedicated depressed participants.

## Materials and methods

This study was conducted in compliance with the Declaration of Helsinki. The Institutional Review Boards of McLean Hospital, Boston University and Williams College approved this study, and all participants provided written informed consent.

### Participants

Twenty-nine individuals with a current diagnosis of MDD, 24 individuals in partial remission from depression, and 35 healthy controls participated in the study. Participants were recruited via fliers posted in public spaces and mental health treatment facilities, as well as via postings in a weekly newsletter for students at a small liberal arts college. Exclusion criteria included history of traumatic brain injury, attention deficit hyperactivity disorder (ADHD), current or past psychosis, pregnancy, current substance or alcohol abuse or dependence, impaired and non-corrected vision. Exclusion criteria were verified through phone call screenings and in-

person interviews and questionnaires, including the Structured Clinical Interview for DSM-4 (SCID) [31]. Visual acuity was assessed with the Freiburg visual acuity test [32].

Depressed participants were diagnosed using the SCID [31], and partially remitted individuals were identified during the SCID if they did not meet current diagnostic criteria for MDD but had met criteria in the past and were currently experiencing subthreshold symptoms, and healthy controls were screened for Axis I disorders with the non-patient version of the SCID. Nineteen participants were taking a psychotropic medication at the time of the study (n = 12 currently depressed, n = 7 remitted depressed).

## Measures, stimuli, and procedure

All participants were required to have 20/40 vision or better, with or without corrective lenses. Participants also completed the word reading subsection of the Wide Range Achievement Test (WRAT; [33]), the Beck Depression Inventory (BDI; [34]) and the Quick Inventory of Depressive Symptomatology-Self Report (QIDS-SR; [35]). Both the QIDS-SR and BDI are frequently used and well-validated. The BDI includes 21 items rated from 0–3, with higher scores indicating greater severity [36]. The QIDS-SR includes 16 items rated from 0–3, with higher scores indicating greater severity [35].

The stimulus was an RDP, 4 degrees in diameter. The dots were 4.3 arcminutes in diameter, moving at a speed of 6.5 degrees per second (Fig 1). The signal to noise ratio, or coherence level, varied across five levels: .04, .08, .16, .32, .64. The coherence level referred to the proportion of dots in the stimulus that were moving in a coherent direction, while the remainder moved randomly. In each trial, a fixation cross was first displayed for 100 msec, followed by the RDP stimulus for 250 msec, followed by a blank screen until the participant responded (Fig 1). A brief stimulus presentation time (250 msec) was chosen to limit the role of eye movements during the task, while being sufficiently long to allow for high accuracy performance in the higher coherence conditions.

Each participant was shown a demonstration of the task and completed a brief round of practice trials before beginning the task to ensure that they understood the instructions. The Baseline condition (the central RDP presented alone) was always presented first, and there were 16 trials for each coherence condition. In the surround condition, following the Baseline condition, the RDP was surrounded by an annulus of moving dots (10 degrees in diameter) moving in the Same or Opposite direction as the central RDP. There was no gap between the central RDP and the surround field, and they did not overlap. The surround field always moved at 100% coherence. Participants completed 8 trials of each condition, for a total of 80 trials, (8 repetitions * 5 coherences * 2 surround conditions, Same or Opposite). Conditions were presented in a random order. The task was to identify the overall direction of motion (left or right) of the central RDP, indicated using the arrow keys on a standard keyboard. Participants' accuracy in each task is represented by the proportion of correct responses out of the total number of trials.

## Outcomes and analyses

CSF was calculated at each coherence level as the difference between Opposite accuracy and Baseline accuracy because improvements in accuracy (i.e., facilitation) tend to occur with a surrounding stimulus that moves in the opposite direction as the center motion [18]. CSS was calculated as the difference between Baseline accuracy and Same accuracy because decreases in accuracy (i.e., suppression) tend to occur when the surround and center move in the same direction [18]. All statistical analyses were performed in R (version 4.0.2). For correlational analyses, Pearson's correlation coefficient was used. To test between-group differences, we

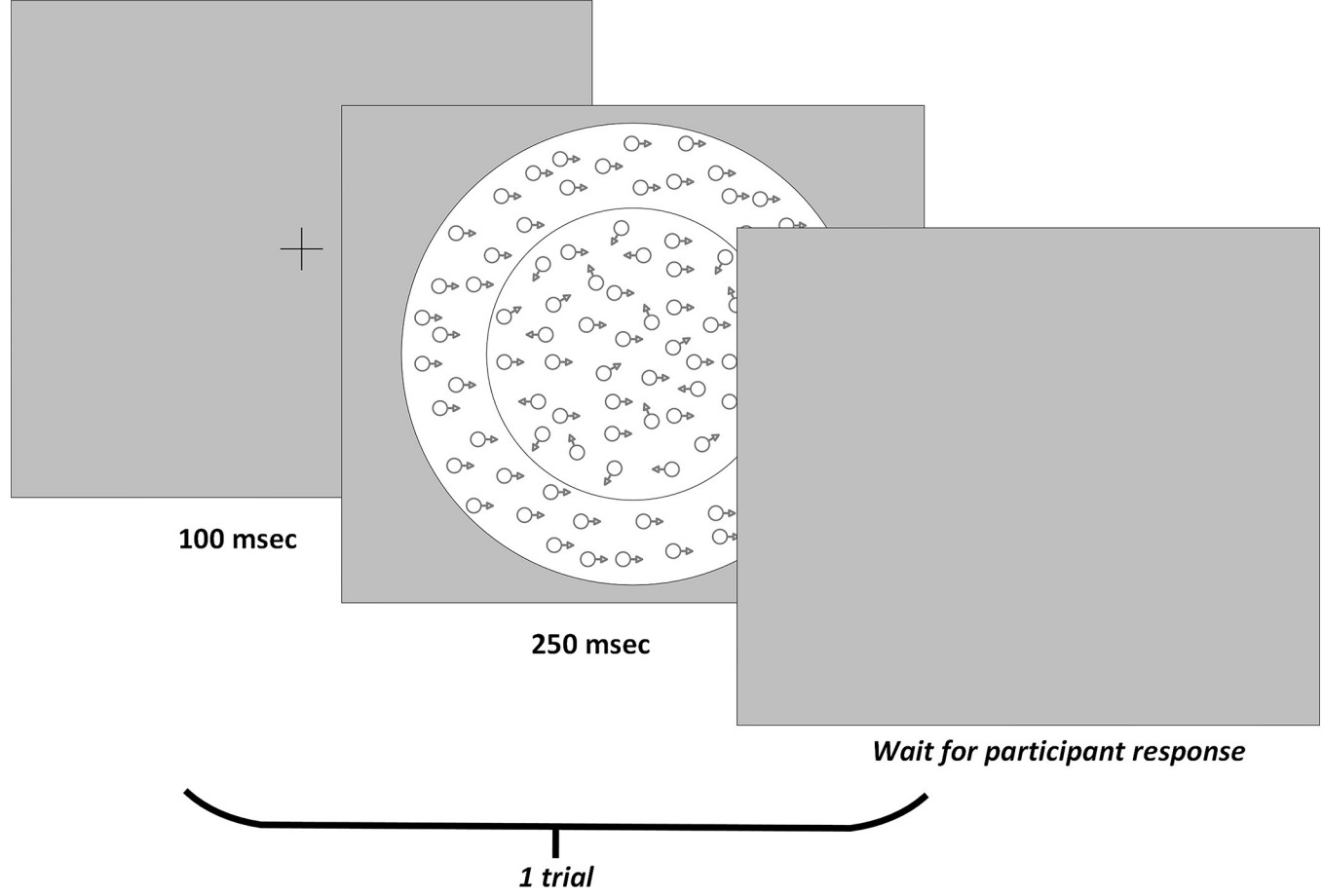

**Fig 1. Random dot pattern stimulus.** This figure illustrates the random dot pattern stimulus used in the present study, including a surrounding annulus, as was used in the surround conditions. This figure is not to-scale.

performed analyses of variance (ANOVAs). We conducted a False Discovery Rate (FDR) correction using the Benjamin-Hochberg procedure [37] with an FDR of 0.10. All p-values reported below remained significant with the FDR correction.

## Results

Of the original sample, 3 participants were excluded due to missing data in some or all psychophysical tasks (2 healthy, 1 currently depressed). Nine additional participants (5 healthy, 2 current, 2 in partial remission) were excluded because their accuracy levels in the 64% coherence condition of the baseline task were 2 standard deviations or more below the mean (M = 0.91, SD = 0.13). Poor performance in this condition may indicate inattention or a lack of understanding. Lastly, we calculated the surround determination metric $D$ for each participant with the following formula:

$$D = \frac{\bar{o} + (1 - \bar{s})}{2}$$

where $\bar{o}$ is average accuracy across all coherence levels in the opposite condition, and $\bar{s}$ is the average accuracy across all coherence levels in the same condition. Two participants (both remitted) were excluded because their surround determination was over 2 standard deviations

**Table 1. Demographic & clinical information.**

| | Current | | Remitted | | Healthy | C v. R | R v. H |
|---|---|---|---|---|---|---|---|
| | *n* | *Mean (SD)* | *n* | *Mean (SD)* | *n* | *Mean (SD)* | *p* | *p* |
| Age Mean | 26 | 35.38 (13.27) | 20 | 39.10 (15.85) | 28 | 34.20 | 0.284 | 0.403 |
| Gender % female | 26 | 65.39 | 20 | 75.00 | 28 | 57.14 | 0.202 | 0.488 |
| BDI | 26 | 31.00 (8.62) | 19 | 15.05 (11.77) | 28 | 1.11 (1.73) | < .001 | < .001 |
| QIDS-SR | 25 | 14.04 (3.27) | 19 | 5.63 (2.39) | 25 | 1.64 (1.70) | < .001 | < .001 |
| WRAT | 25 | 49.96 (3.76) | 20 | 51.30 (2.58) | 22 | 50.64 (4.11) | 0.164 | 0.531 |

C = Current; R = Remitted; H = Healthy

greater or less than the mean (M = 0.60, SD = 0.12). Following these exclusions, the final sample of 74 adults included 26 current (including 12 medicated), 20 remitted (including 7 medicated), and 28 healthy. A summary of the sample's demographics and clinical data is in Table 1. There were significant between-group differences on the BDI and QIDs, but not on the WRAT.

In the following analyses, the current and remitted groups were combined into one "depressed" group and compared to the healthy controls. We chose to combine the currently depressed and partially remitted participants because all partially remitted participants responded to an advertisement recruiting people with depression, indicating that they all self-identified as depressed, and they endorsed some current symptoms of depression. Furthermore, their BDI scores were substantially greater than those of the healthy group, and most of the remitted group had BDI scores consistent with mild, moderate, or severe depression (Fig 2; [34]). There were no differences between the partially remitted and currently depressed participants on accuracy in the Baseline, Same, or Opposite conditions, or in facilitation or suppression effects (all p-values >0.3).

We ran all analyses in two ways: first, including both medicated and unmedicated participants (n = 46 depressed, 28 healthy), and second, including only unmedicated participants (n = 27 depressed, 28 healthy). When the homoscedasticity requirement was not met, we used Greenhouse-Geisser estimation. Shapiro-Wilk tests indicated that the data was not normal, but we moved forward with our planned ANOVAs because recent statistical studies have shown that Type I error in ANOVAs is robust to nonnormality [38, 39].

### Baseline accuracy

**Including medicated and unmedicated participants.** A two-way repeated measures ANOVA compared baseline accuracy levels by group and coherence level. There was no main effect of group for baseline accuracy between depressed and healthy participants (F(1, 72) = 0.206, p = 0.651, $\eta^2 = .003$, 90% $\eta^2$CI [0.00, 0.053]) and there was no significant group by coherence interaction (F(3.45, 248.66) = 1.467, p = .212, $\eta^2 = 0.02$, 90% $\eta^2$CI [0.00, 0.045]), but there was a significant main effect of coherence (F(3.45, 248.66) = 164.602, p < .001, $\eta^2 = 0.696$, 90% $\eta^2$CI [0.643, 0.730]; Fig 3A), with significantly greater accuracy at higher coherence levels.

**Including only unmedicated participants.** There was no main effect of group for baseline accuracy between unmedicated depressed and healthy participants (F(1, 53) = 0.041, p = 0.840, $\eta^2 = .001$, 90% $\eta^2$CI [0.00, 0.041]) and there was no significant group by coherence interaction (F(3.37, 178.79) = 0.786, p = .516, $\eta^2 = 0.015$, 90% $\eta^2$CI [0.00, 0.038]), but there was a significant main effect of coherence (F(3.37, 178.79) = 139.240, p < .001, $\eta^2 = 0.724$, 90% $\eta^2$CI [0.665, 0.760]; Fig 3B), with significantly greater accuracy at higher coherence levels.

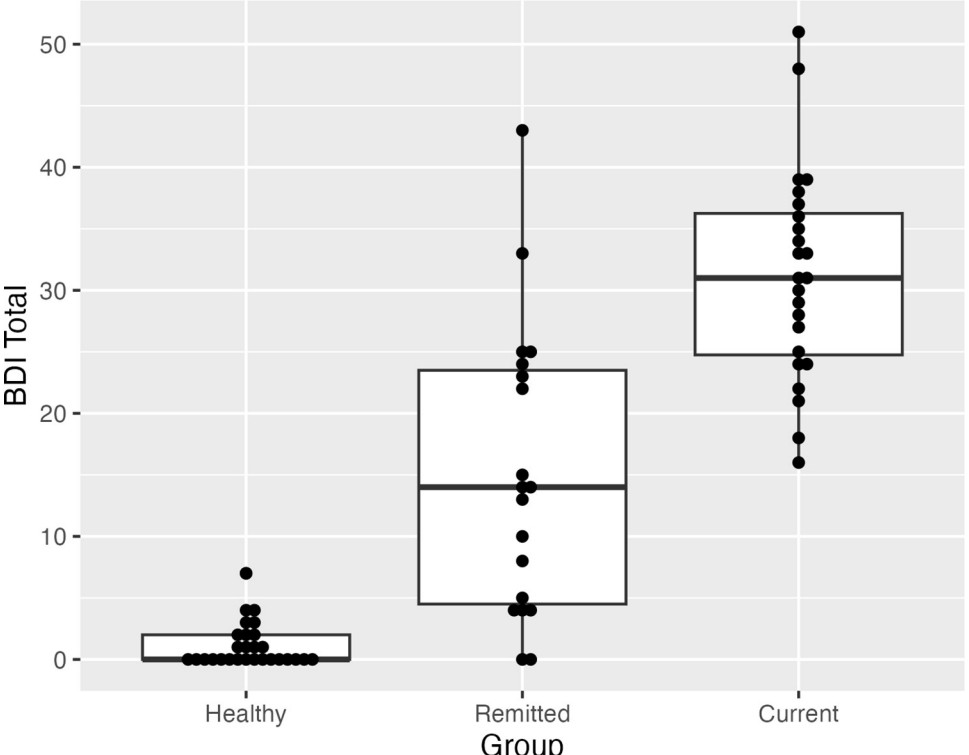

**Fig 2. Average total BDI.** This figure illustrates a box plot of total BDI score in the healthy control, remitted depressed, and currently depressed groups.

## Surround condition accuracy

**Including medicated and unmedicated participants.** A three-way repeated measures ANOVA compared accuracy levels by group (depressed vs. healthy), surround direction (Same vs. Opposite), and coherence (Fig 4A). There were significant main effects of surround direction (F(1,72) = 93.803, p < .001, $\eta^2 = 0.566$, 90% $\eta^2$CI [0.435, 0.652]) and coherence (F

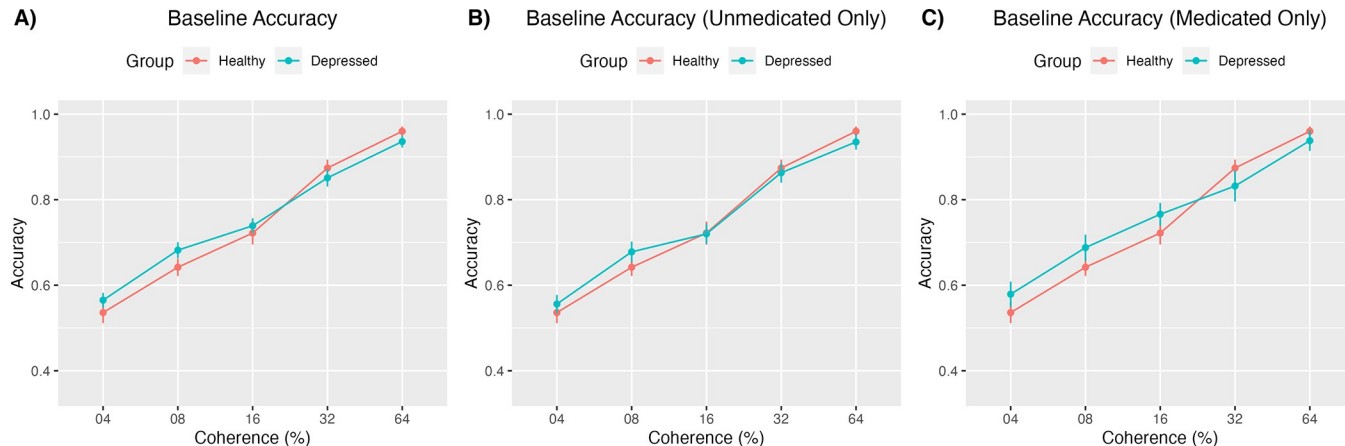

**Fig 3. Accuracy in baseline condition.** This figure depicts accuracy in the baseline condition for the healthy and depressed groups across all 5 coherence levels. Part A (left) includes both medicated and medicated participants, Part B (middle) includes only unmedicated participants, and Part C (right) includes only medicated participants.

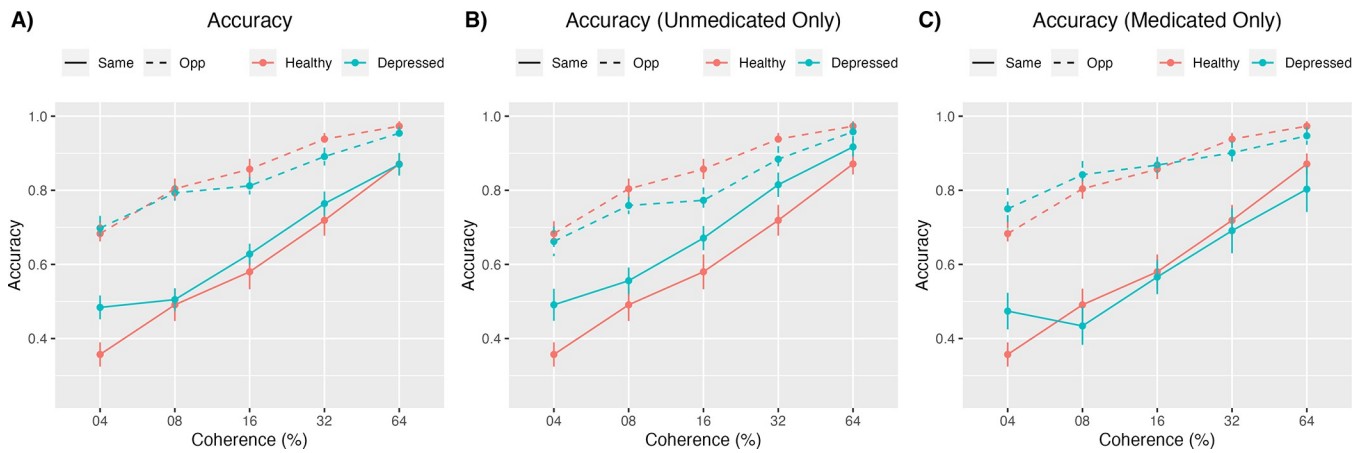

**Fig 4. Accuracy in same and opposite conditions.** This figure depicts accuracy in the same and opposite surround conditions for the healthy and depressed groups across all 5 coherence levels. Part A (left) includes medicated participants, and Part B (right) does not.

(3.21, 231.36) = 117.759, p < .001, $\eta^2 = 0.621$, 90% $\eta^2$CI [0.555, 0.664]), with significantly greater accuracy in the Opposite condition and at higher coherence levels. There was a significant direction by coherence interaction (F(4, 288) = 11.506, p < .001, $\eta^2 = 0.138$, 90% $\eta^2$CI [0.073, 0.190]). There was no significant main effect of group (F(1,72) = .392, p = .533, $\eta^2 = 0.005$, 90% $\eta^2$CI [0.00, 0.064]), group by direction interaction (F(1,72) = 2.335, p = .131, $\eta^2 = 0.031$, 90% $\eta^2$CI [0.00, 0.120]), group by coherence interaction (F (3.21, 231.36) = 1.565, p = 0.196, $\eta^2 = 0.021$, 90% $\eta^2$CI [0.00, 0.049]), or three-way interaction (F(4,288) = 0.779, p = 0.54, $\eta^2 = 0.011$, 90% $\eta^2$CI [0.00, 0.024]).

**Including only unmedicated participants.** A three-way repeated measures ANOVA compared accuracy levels by group, direction, and coherence (Fig 4B). There were significant main effects of direction (F(1,53) = 66.403, p < .001, $\eta^2 = 0.556$, 90% $\eta^2$CI [0.397, 0.655]) and coherence (F(3.17, 167.78) = 112.374, p < .001, $\eta^2 = 0.680$, 90% $\eta^2$CI [0.611, 0.723]), with significantly greater accuracy in the Opposite condition and at higher coherence levels. There was a significant group by direction interaction (F(1,53) = 8.389, p = .0005, $\eta^2 = 0.137$, 90% $\eta^2$CI [0.0247, 0.277]) and a significant direction by coherence interaction (F(4, 212) = 8.375, p < .001, $\eta^2 = 0.136$, 90% $\eta^2$CI [0.061, 0.195]). There was no significant main effect of group (F(1,53) = .947, p = .335, $\eta^2 = 0.018$, 90% $\eta^2$CI [0.00, 0.111]), group by coherence interaction (F(3.17, 167.78) = 0.528, p = 0.673, $\eta^2 = 0.010$, 90% $\eta^2$CI [0.00, 0.029]), or three-way interaction (F(4,212) = 0.722, p = 0.577, $\eta^2 = 0.013$, 90% $\eta^2$CI [0.00, 0.030]). We conducted post-hoc t-tests to reveal the nature of the group by direction interaction. The depressed group had significantly greater accuracy than the healthy group in the Same condition ($M_{diff}$ = 0.09, t(270.47) = -2.81, p = 0.01, d = 0.34, 95% CI [0.10, 0.58]), but not the Opposite condition ($M_{diff}$ = -0.04, t(263.15) = 1.99, p = 0.05, d = -0.24, 95% CI [-0.48, 0.00]).

## Facilitation and suppression effects

**Including medicated and unmedicated participants.** Two-way ANOVAs compared facilitation and suppression effects by group and coherence (Fig 5A). For facilitation, there was a significant main effect of coherence (F(3.09, 222.6) = 7.490, p < .001, $\eta^2 = 0.094$, 90% $\eta^2$CI [0.034, 0.149]) with greater facilitation at lower coherence levels. There was no significant main effect of group (F(1,72) = 1.787, p = .186,

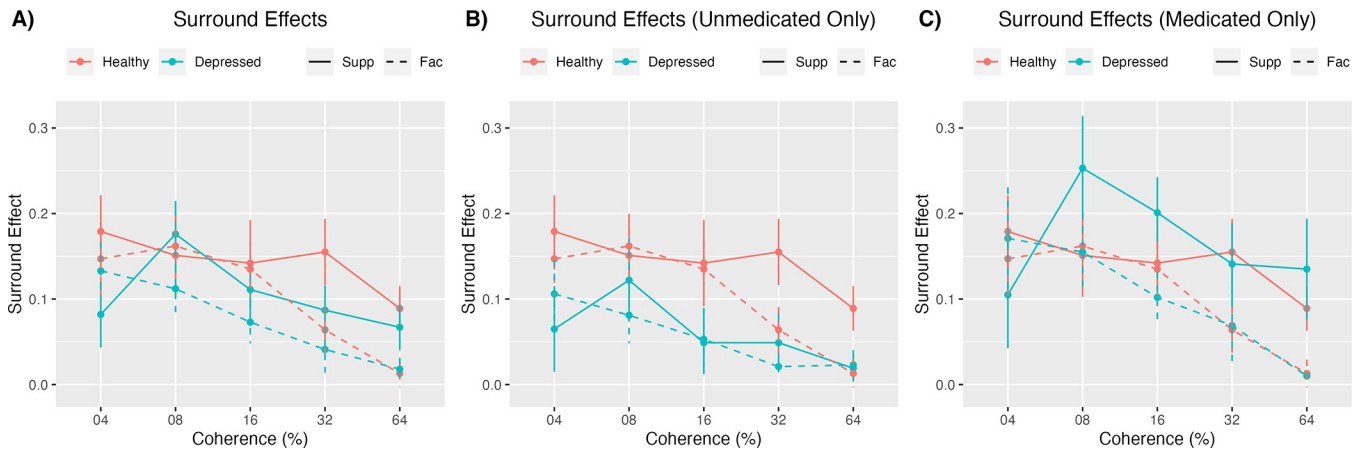

**Fig 5. Suppression and facilitation effects.** This figure depicts the suppression and facilitation effects for the healthy and depressed groups across all coherence levels. Part A (left) includes medicated participants, and Part B (right) does not. Suppression and facilitation effects likely decrease with greater coherence due to reduced difficulty of the task at higher coherence, which reduces variability of participant responses and the ability of the surround to influence responses.

$\eta^2 = 0.024,\ 90\%\ \eta^2\text{CI}\ [0.00,\ 0.108])$ or group by coherence interaction ($F(3.09, 222.6) = 0.454$, p = 0.721, $\eta^2 = 0.006,\ 90\%\ \eta^2\text{CI}\ [0.00,\ 0.020]$). For suppression, there was no significant main effect of group ($F(1,72) = 1.272$, p = .263, $\eta^2 = 0.017,\ 90\%\ \eta^2\text{CI}\ [0.00,\ 0.094]$), main effect of coherence ($F(3.59, 258.83) = 1.752$, p = 0.146, $\eta^2 = 0.024,\ 90\%\ \eta^2\text{CI}\ [0.00,\ 0.050]$), or group by coherence interaction ($F(3.59, 258.83) = 1.028$, p = 0.389, $\eta^2 = 0.014,\ 90\%\ \eta^2\text{CI}\ [0.00,\ 0.033]$).

**Including only unmedicated participants.** Two-way ANOVAs compared facilitation and suppression effects by group and coherence (Fig 5B). For facilitation, there was a significant main effect of coherence ($F(3.09, 163.51) = 4.983$, p = .002, $\eta^2 = 0.086,\ 90\%\ \eta^2\text{CI}\ [0.020,\ 0.147]$) with greater facilitation at higher coherence levels. There was no significant main effect of group ($F(1,53) = 3.776$, p = .057, $\eta^2 = 0.067,\ 90\%\ \eta^2\text{CI}\ [0.00,\ 0.191]$) or group by coherence interaction ($F(3.09, 163.51) = 0.744$, p = 0.531, $\eta^2 = 0.014,\ 90\%\ \eta^2\text{CI}\ [0.00,\ 0.039]$). For suppression, there was a main effect of group ($F(1,53) = 5.420$, p = .024, $\eta^2 = 0.093,\ 90\%\ \eta^2\text{CI}\ [0.007,\ 0.226]$), with depressed individuals showing less suppression than controls (Fig 5B). There was no significant main effect of coherence ($F(3.55, 188.17) = 1.524$, p = 0.203, $\eta^2 = 0.028,\ 90\%\ \eta^2\text{CI}\ [0.00,\ 0.061]$) or group by coherence interaction ($F(3.55, 188.17) = 0.459$, p = 0.744, $\eta^2 = 0.009,\ 90\%\ \eta^2\text{CI}\ [0.00,\ 0.022]$).

## Correlations

For each participant, we calculated the threshold of coherence at which they could accurately detect the direction of motion. These thresholds were calculated by fitting their data to a Weibull function: $y = 1 - .5*e^{\left(\frac{-x}{a}\right)^b}$ where $y$ is the proportion of trials judged correctly, $x$ is the coherence of the RDP, and $a$ and $b$ are two curve-fitting parameters [40]. For the Same condition, for participants who did not reach 80% accuracy at any coherence level, we used .64, the highest possible coherence in the task. We tested whether these thresholds were correlated with depression symptom severity; results are summarized in Table 2. For all conditions (Baseline, Same, and Opposite) there were no significant correlations between accuracy threshold and BDI score or QIDS score.

**Table 2. Correlations.**

| | Baseline Threshold | | Same Threshold | | Opposite Threshold | |
|---|---|---|---|---|---|---|
| | r (p) | DF | r (p) | DF | r (p) | DF |
| BDI | -0.008 (0.946) | 71 | -0.091 (0.345) | 71 | 0.148 (0.218) | 71 |
| QIDS | -0.058 (0.645) | 66 | -0.042 (0.741) | 66 | 0.069 (0.580) | 66 |

DF = degrees of freedom

## Discussion

The present study was the first to examine performance on a global motion perception task in depressed individuals, showing normal baseline performance using a standard RDP stimulus, but abnormally reduced contextual effects when surrounding that stimulus with an annulus. The starkest group difference was the unmedicated depressed group's enhanced performance when perceiving the direction of motion in the Same condition, suggesting reduced CSS.

Though prior studies had examined related visual abilities in depression, such as discriminating the direction of motion in grating stimuli [10–12] or detecting contrast [41], the present study was the first to investigate global motion processing in depressed individuals. Global motion processing relies on distinct neural mechanisms from other visual tasks that have been studied in depression [42]. For example, contrast detection begins to be processed in the retina itself, due to the high contrast sensitivity of retinal ganglion cells [43], and direction discrimination based on local signals can occur in early stages of cortical processing, including area V1, while global motion processing occurs in downstream areas of the visual stream, including MT [44]. Therefore, the lack of significant difference in baseline accuracy between depressed and healthy participants in the present study is a novel result that supports the integrity of global motion processing in depression. Two prior studies [10, 11] had both shown normal local motion processing of small, high contrast stimuli. The present study is consistent with those results, extending them to the domain of global motion processing. One recent study by Song et al. [12] did show reduced performance in depressed individuals judging the direction of small, high contrast motion stimuli (local motion). It is possible that the high percentage of patients taking antipsychotic medications in Song et al.'s sample (49%) contributed to the baseline deficit in motion perception, given that antipsychotic medications are known to affect performance on many kinds of cognitive tests [45]. Furthermore, in-tact global motion processing in depression in the present study suggests that the observed differences in CSS in the present study are not due to a generalized cognitive deficit, or to impaired motion processing in general [46].

The present study's finding of reduced CSS in depression is consistent with prior research showing abnormal inhibitory effects within motion processing in depressed or remitted-depressed individuals [10–12]. The present findings extend the pattern of abnormal results in depression to the domain of global motion processing. Golomb et al. [11] found that unmedicated participants who were fully remitted from depression demonstrated significantly lower duration thresholds, i.e., improved performance compared with healthy controls when perceiving the direction of motion of a large grating stimulus. There was no significant difference with a small grating stimulus in the same participants. In the present study and in Golomb's study, the higher accuracy in depressed individuals for the conditions where CSS normally hampers perception may have been driven by reduced CSS effects in the depressed groups. Song et al. [12] reported reduced suppression indices in currently-depressed individuals using a local motion task, although these were driven by impaired performance in the small

condition, combined with normal performance in the large condition (where CSS exerts its effects; [16]). Regardless, the fact that performance was not also impaired in the large condition suggests that CSS was reduced in the depressed individuals in that study. The other study on inhibitory processing within motion perception in depression [10] showed *increased* CSS effects in currently depressed individuals at high contrast, and non-significantly reduced CSS effects at low contrast, both peaking at presentation times around 66 msec. The present results of reduced inhibitory effects in global motion contrast with those, especially since there is partial overlap between the present sample and the one comprising that study. It does appear that stimulus features such as contrast, presentation time, and the specific nature of the task affect how alterations in depressed brains play out in terms of visual motion inhibition.

The medication-related findings in the present study and Norton et al. [10] suggest that psychotropic medication may increase the suppression effect at multiple stages of visual motion processing. In the present study, accuracy was higher in unmedicated depressed subjects compared to controls in the Same surround condition, indicating decreased suppression, however it did not differ from controls when adding the medicated subjects to the analysis. This suggests that medication normalized, or increased, CSS. We conducted additional analyses to verify differences by medication status within the depressed group; please see the supporting information [S1 Text, S1 and S2 Figs]. Similarly, Norton et al.'s [10] depressed sample was evenly split between medicated and unmedicated participants. The medicated participants in that study also had greater CSS than the unmedicated participants (difference only significant at lower contrast). It is possible that in Song et al.'s [12] depressed sample, CSS was increased by SSRI medication (taken by over 90% of the depressed sample in that study), which may partially explain why they did not find enhanced visual perception in the depressed sample as Golomb and colleagues found using the same task in unmedicated, remitted-depressed individuals.

While altered inhibitory processing in a variety of motion domains is emerging as a robust result in depression, the causal role of altered GABA levels in that pattern is weaker than once thought. A meta-analysis of seven studies showed no significant difference in GABA concentration in the occipital cortices of depressed and healthy participants [47, 48], calling into question a major piece of the assumption that alterations in GABA concentrations could underly the altered inhibitory effects in the perceptual performance of depressed or recovered-depressed individuals. In addition, in the one extant study where both GABA and suppression scores were measured, the two variables were not correlated in depressed participants (though healthy controls in that study did show such a correlation; [12]). Another study found that alcohol administration did not lead to an increased suppression index, which would be expected given alcohol's potentiation of GABA-ergic transmission [49].

Further, from the side of basic visual research, GABA's role in surround suppression has been questioned as well. First, local blockage of GABA inputs does not disrupt motion suppression [44, 50]. In addition, a recent theory proposes that surround suppression of various types of motion signals can be explained by the stabilized supralinear network (SSN; [48]). The SSN model suggests that, for strong stimuli, neuronal gain (i.e., the change in electrical output per input) is enhanced by network activation, leading to supralinear activation behavior where two inputs would drive greater activity than the sum of either alone. As the input strength increases, the output gets stronger and stronger until a breaking point is reached, at which point the outputs of individual units sum sublinearly due to emergent properties of the network dynamics [51]. In this model, GABA is not implicated in CSS because the suppression is due to a relative lack of excitatory input compared with the suppressive input that arises from strong stimuli, rather than an inhibitory neurotransmitter. Whether the SSN can account for differences in CSS between depressed and healthy individuals remains an open question. It is

possible that subtle changes in network dynamics of the supralinear or sublinear portions of neuronal response to strong and weak stimuli can explain some of the results thus far. For example, a higher breaking point before the system switches from supralinear to sublinear behavior, or weaker gain for neuronal response as external input strength increases, could explain the reduced CSS effects shown in this and prior studies. Given that one study found increased CSS effects in some conditions, and trends towards reduced ones in others [10], this will be an interesting topic to test and explore in different aspects of the visual system.

Schallmo et al. [52] have proposed an additional model of suppression and facilitation that does not rely principally on GABAergic inhibition. Their theory makes use of divisive normalization, the idea that neuronal output depends not only on the input, but also on the activity of surrounding neurons [52]. They propose a model of neuronal response based on the ratio between an excitatory drive, defined by the strength of the input (i.e., contrast level), and a suppressive drive [52]. This model can explain both CSF and CSS, eliminating the need for separate neural mechanisms for facilitatory and suppressive behavior (see [52] for a full explanation). However, this model does not explain the results of the present study, which indicate differences in CSS, but not CSF, in unmedicated depressed and healthy adults. Future research is needed to determine if the results of the present study are evidence of distinct neural mechanisms of CSS and CSF, or if alterations in the relative strength of suppressive or faciliatory contextual inputs in depression can be explained through the divisive normalization model.

Notably, other research has suggested that depressed individuals have significantly shorter saccades than controls in eye movement tests [53], which could be related to group differences in RDP direction of motion performance. However, if this were the case, we would expect to see differences in the Baseline condition, as well as the Same and Opposite conditions. Because this study only found differences in the Same condition, we do not believe saccade length is implicated in the depressed group's performance.

Overall, future research is needed to clarify which aspects of inhibitory processing within the visual system are increased or decreased in recovered or active states of depression, as well as more basic research on the neural mechanisms underlying this process in the healthy brain. One limitation of the present study is the binocular assessment of visual acuity. Research suggests that surround suppression is lower for monocular than binocular vision [54], and future work should assess visual acuity monocularly as participants with an amblyopic eye may demonstrate lower CSS. Future work may also test the effects of specific psychotropic medications on CSS in depression in randomized controlled trials, as the naturalistic nature of the medication data in the present study prevents any causal interpretation. An especially important future direction is investigating the effect of GABAergic medications on CSS in healthy and depressed individuals. The mechanisms behind CSS remain an open question, and the medication-related effects observed in the present study may aid in justifying future research to clarify the role of medication and GABA concentration in CSS.

## Conclusion

The present study's results suggest that abnormal visual perception related to CSS effects in depressed individuals is not limited to local motion processing assessed with uniform grating stimuli, but extends into global motion processing assessed with RDPs as well. Psychotropic medication may separately increase CSS, although the mechanism for this increase is unclear due to recent research challenging the GABA hypothesis. Abnormal inhibitory processing in visual perception may become a reliable marker of depression, allowing for earlier identification and intervention. Additional research is needed to identify potential differences between

currently and previously depressed individuals, and between medicated and unmedicated individuals.

## Supporting information

**S1 Text. This supporting information compares accuracy, CSS, and CSF by medication status within the depressed group.**
(DOCX)

**S1 Fig. Baseline accuracy in medicated and unmedicated depressed participants.** This figure depicts accuracy among medicated vs. unmedicated participants within the depressed group across all coherence levels.
(TIF)

**S2 Fig. Accuracy and surround effects in medicated and unmedicated depressed participants.** This figure depicts accuracy in the same and opposite conditions (Part A, left) and suppression and facilitation effects (Part B, right) among medicated vs. unmedicated participants within the depressed group across all coherence levels.
(TIF)

**S1 Data.**
(XLSX)

## Acknowledgments

We would like to acknowledge Diana Matthiessen, Juna Khang, and Ziqing Zong for their contributions to data collection at Williams College. This paper is dedicated to the memory of Yue Chen.

## Author Contributions

**Conceptualization:** Grace E. Murray, Daniel J. Norton.

**Data curation:** Grace E. Murray, Daniel J. Norton.

**Formal analysis:** Grace E. Murray.

**Funding acquisition:** Daniel J. Norton.

**Investigation:** Daniel J. Norton.

**Methodology:** Daniel J. Norton.

**Project administration:** Grace E. Murray, Daniel J. Norton.

**Resources:** Daniel J. Norton.

**Software:** Grace E. Murray, Daniel J. Norton.

**Supervision:** Daniel J. Norton.

**Visualization:** Grace E. Murray.

**Writing – original draft:** Grace E. Murray, Daniel J. Norton.

**Writing – review & editing:** Grace E. Murray, Daniel J. Norton.

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
