## [Decision Letter · Decision Letter 0]

29 May 2023

PONE-D-23-03585Reduced visual context effects in global motion processing in depressionPLOS ONE

Dear Dr. Norton,

Thank you for submitting your manuscript to PLOS ONE. After careful consideration, we feel that it has merit but does not fully meet PLOS ONE’s publication criteria as it currently stands. Therefore, we invite you to submit a revised version of the manuscript that addresses the points raised during the review process.

We look forward to receiving your revised manuscript.

Kind regards,

Giulia Prete

Academic Editor

PLOS ONE

Journal Requirements:

"This work was supported by a grant from McLean Hospital and the Rossano family to DJN, as well funds from Gordon and Williams College. This research was also supported in part by NINDS F31 NS07682 to DJN."

"This work was supported by a grant from McLean Hospital and the Rossano family to DJN, as well funds from Gordon and Williams College. This research was also supported in part by NINDS F31 NS07682 to DJN."

Reviewers' comments:

Reviewer's Responses to Questions

**Comments to the Author**

1. Is the manuscript technically sound, and do the data support the conclusions?

Reviewer #1: Yes

Reviewer #2: Yes

2. Has the statistical analysis been performed appropriately and rigorously? 

Reviewer #1: Yes

Reviewer #2: Yes

3. Have the authors made all data underlying the findings in their manuscript fully available?

Reviewer #1: No

Reviewer #2: No

4. Is the manuscript presented in an intelligible fashion and written in standard English?

Reviewer #1: Yes

Reviewer #2: Yes

5. Review Comments to the Author

Reviewer #1: This investigation explores center-surround interactions (facilitations and suppression) in global motion processing in adults with depression using random dot patterns in a motion direction discrimination task. They also compared medicated and non-medicated participants with depression. Results show that in general, global surround suppression and facilitation is not affected in depressed participants compared to controls, however, when a group of non-medicated depressed participants were selected, they presented a reduced center-surround suppression compared to controls. It seems that psychotropic medications mitigated the reduced abnormal suppression.

I find this to be an interesting and well written paper, however I have a few comments and concerns.

The authors state that this is the first study to examine global motion perception in depressed individuals. I do not agree with the definition of local and global motion given by the authors. The use of gratings doesn’t necessarily mean local motion, and the use of random dots doesn’t mean global motion, it depends basically on the size of the stimulus and the activated brain areas (e.g. V1 is considered local and MT global). It is also well known that center-surround MT neurons respond more strongly to a moving large low-contrast stimuli than to one of high contrast. Physiological studies have shown that these MT cells are tuned to the size of the stimulus and suppress the response when the high contrast stimulus is larger than the neuron’s classical receptive field (Pack et al. 2005). Psychophysical studies have also shown that duration thresholds increase with increasing size for moving high contrast gratings and decrease with increasing size for low contrast ones, suggesting a perceptual correlate of the physiological center-surround suppression (Tadin et al. 2003). Therefore, the studies (cited by the authors) testing depressive participants where they use large gratings are also examining global motion perception.

One question I had reading the manuscript is how the authors tested the vision of the participants. Did they measure the spatial acuity for each eye separately? What instrument did they use? Did they measure the stereoacuity? This is an interesting point given that a recent study has shown that surround suppression is lower for monocular than binocular vision (Arranz-Paraíso et al. 2021). Thus, participants with an amblyopic eye could show lower suppression. Interestingly, ketamine that is used as an antidepressant in humans, is effective in treating adult amblyopia (Grieco et al., 2020). I wonder whether small eye deviations are more present in unmedicated depressed participants or if they present poorer stereoacuity. I think the authors should discuss this aspect in the paper.

Another recent study has shown differences in eye movements between MDD and controls (Takahashi et al, 2021). Is it possible that these differences in eye movements could explain these results for the unmedicated participants?

About the analysis, I would like to see in the main document the analysis including only the medicated participants compared with the healthy controls. One figure of accuracy and one of surround effects comparing controls and medicated participants would be very informative (e.g. Figures 4C and 5C).

Why is the surround effect reduced with increasing coherence? Significant for facilitation but for suppression not significant if we take all coherences. I guess that if the authors fit regression lines to the data of Figures 5A and B, theses would be negative and probably significant. For suppression, the differences between 0.8% and 64% are clear, I would like the authors discuss this point.

About the representation of the data, I think it is very informative to use beeswarm plot with boxplots to show whether there are outliers in the conditions.

Minor

-Regarding trials, although there is an image in Figure 1 describing them, I would like the authors to include a description of the trials in the main text. Why the presentation time is 250 msec? Have the authors tested other durations? The classical way to test motion surround suppression is measuring duration thresholds. Those are much lower than the duration used here.

-Page 9. How is the accuracy measured? Is it percentage of correct responses?

-Page 10. “…26 current (including 12 medicated), 20 remitted (including 9 medicated),…”

This is 26+20 =46 depressed participants. Thus, 46-(12+9)=25 unmedicated. In page 11 the authors say 27 unmedicated and also in the abstract.

-Page 11. For the ANOVAs. Did the authors test the normality of the data?

-Page 14. What is "higher facilitation"? Do you mean “lower Surround effect”?

-Page 14-15. Are there differences between facilitation and suppression for the groups of participants tested?

-Page 17. When the authors describe Golomb’s study, I think is better to talk about duration thresholds instead of accuracy.

-Check the authors in the reference 19.

References

Arranz-Paraíso, S., Read, J.C.A. & Serrano-Pedraza, I. (2021). Reduced surround suppression in monocular motion perception. Journal of Vision;21(1):10

Grieco SF, Qiao X, Zheng X, Liu Y, Chen L, Zhang H, Yu Z, Gavornik JP, Lai C, Gandhi SP, Holmes TC, Xu X. Subanesthetic Ketamine Reactivates Adult Cortical Plasticity to Restore Vision from Amblyopia. Curr Biol. 2020 Sep 21;30(18):3591-3603.e8. doi: 10.1016/j.cub.2020.07.008.

Pack CC, Hunter JN, Born RT. Contrast dependence of suppressive influences in cortical area MT of alert macaque. J Neurophysiol. 2005 Mar;93(3):1809-15. doi: 10.1152/jn.00629.2004. Epub 2004 Oct 13. PMID: 15483068.

Takahashi J, Hirano Y, Miura K, Morita K, Fujimoto M, Yamamori H, Yasuda Y, Kudo N, Shishido E, Okazaki K, Shiino T, Nakao T, Kasai K, Hashimoto R and Onitsuka T (2021) Eye Movement Abnormalities in Major Depressive Disorder. Front. Psychiatry 12:673443

Reviewer #2: In the manuscript entitle ‘Reduced visual context effects in global motion processing in depression’, Murray & Norton study visual motion perception in patients with depression. They find altered motion processing in depression, but only on those patients without medication. This is a nice addition to the literature on visual abnormalities in depression. The manuscript is well-written, methods are technically sound, and results are clearly presented. I have only a few minor comments, presented below.

1) How the exclusion criteria were checked? Only subjective report or other means, please clarify.

2) Why the analysis were repeated separately with medicated and unmedicated? The other option would have been medication as an additional between-subjects factor. Was there any reason for separate analyses?

3) The authors could consider linear mixed effects model instead of repeated measures ANOVA. Some of the excluded participants could be possible added to the analysis, since the mixed model can model data with missing values. The linear mixed effects model also has larger statistical power than rmANOVA. And perhaps the total number of statistical tests in the manuscript (and supplement) could also be reduced with a little bit more complex statistical model.

4) Quite a few ANOVAs were conducted. Did the p-values remain significant after correction for multiple comparisons? For example, FDR-correction across all the reported p-values would be simple to calculate.

5) In discussion, the SSN model is presented. What then would be the cause of abnormal motion perception in depression, altered neuronal gain or network dynamics or something else? Could the authors speculate a bit more based on the SSN model?

6. PLOS authors have the option to publish the peer review history of their article (what does this mean?). If published, this will include your full peer review and any attached files.

Reviewer #1: No

Reviewer #2: No

---

## [Author Response · Author response to Decision Letter 0]

27 Jul 2023

Reviewer #1: 

This investigation explores center-surround interactions (facilitations and suppression) in global motion processing in adults with depression using random dot patterns in a motion direction discrimination task. They also compared medicated and non-medicated participants with depression. Results show that in general, global surround suppression and facilitation is not affected in depressed participants compared to controls, however, when a group of non-medicated depressed participants were selected, they presented a reduced center-surround suppression compared to controls. It seems that psychotropic medications mitigated the reduced abnormal suppression.

I find this to be an interesting and well written paper, however I have a few comments and concerns.

Response: We thank the reviewer for his/her/their encouraging comments and are pleased they find ours to be an interesting study. 

The authors state that this is the first study to examine global motion perception in depressed individuals. I do not agree with the definition of local and global motion given by the authors. The use of gratings doesn’t necessarily mean local motion, and the use of random dots doesn’t mean global motion, it depends basically on the size of the stimulus and the activated brain areas (e.g. V1 is considered local and MT global). It is also well known that center-surround MT neurons respond more strongly to a moving large low-contrast stimuli than to one of high contrast. Physiological studies have shown that these MT cells are tuned to the size of the stimulus and suppress the response when the high contrast stimulus is larger than the neuron’s classical receptive field (Pack et al. 2005). Psychophysical studies have also shown that duration thresholds increase with increasing size for moving high contrast gratings and decrease with increasing size for low contrast ones, suggesting a perceptual correlate of the physiological center-surround suppression (Tadin et al. 2003). Therefore, the studies (cited by the authors) testing depressive participants where they use large gratings are also examining global motion perception.

Response: We appreciate the reviewer’s thoughts on the neurophysiological correlates of local and global motion processing, especially that grating stimuli, while simple, do activate MT neurons that sum motion over relatively large areas. We still feel that for the purposes of this paper, the way we distinguish global versus local is preferable, though we have clarified what the distinction we are making is, and the rationale for using it. There are three reasons for our decision detailed below. 

 While the reviewer’s points have helped us clarify our distinction, we feel that our use of the local/global distinction is consistent with prior literature in basic psychophysical [21], neurophysiological [28] and clinical studies [20]. See addition below for examples. 

 Defining the sort of motion in a stimulus based on the brain areas it activates introduces some problems. For example, in Figure 1a from Pack et al., 2005, we see a single neuron that behaves somewhat locally for high contrast stimuli (having a maximal response for stimuli of about 8 degrees, but more globally for low contrast stimuli (increasing its activity to 25 or 30 degrees). More generally, in MT there are two major types of cells, band cells which integrate motion over space (allowing perception of global motion) and interband cells which behave similarly to V1 motion neurons in that they respond to local motion primarily [28]. Certainly, defining global motion as “that which activates MT neurons” does not seem like the clearest definition, at least for the purposes of the current work.

 It is not necessarily true that prior studies using large grating stimuli in depressed subjects were testing global motion processing. While large gratings would activate MT cells, as well as cells in V1, it is unclear which cells would be most important for completing the task in a human. To our knowledge, that question remains unanswered, but there are reasons to believe that neurons responsible for processing lower-level features may correspond more closely to the perceptual behavior (i.e., judgement) in relevant cases. For example, MT neurons are exceptionally sensitive (on par with the sensitivity of the monkey in which they reside) for judging the direction of large stimuli that are corrupted by noise (i.e., random dot patterns). They are, however, less sensitive and less aligned with the sensitivity of the monkey in which they dwell on tasks that have a uniform, noiseless stimulus but require fine distinctions (e.g., gratings; [18]). Together, these findings suggest that although MT neurons are activated when viewing a large grating stimulus, it is possible that are not critical for generating perceptual responses in tasks using grating stimuli. We increased the precision of our wording when discussing what prior studies were limited to (first sentence in final paragraph of the addition below).

Revision: Page 5, “Prior basic and clinical research has distinguished between motion processing as being local or global in nature [19–21]. For example, the receptive field of particular motion processing cells in V1 or MT may have access to only the “local motion” of a part of an object but not the whole object itself, the “global motion” [18]. Or, in the case of a random dot pattern, there may be many local motions, many of which are random, while the direction of global coherent motion is still apparent [19], forcing the perceiving organism to use global motion signals to perform a task. This distinction of global versus local motion has also been applied in the realm of brain disorders, where one study showed that direction discrimination was impaired in global, but not local motion in schizophrenia [20]. In the present study, we refer to global motion as that which computationally requires the integration of multiple local motion signals to correctly perform the task. Local motion is that which can be correctly perceived by relying on any local receptive field, as in a grating stimulus. Existing research regarding CSS in depression has been limited to studies using grating stimuli, local motion tasks [10–12], and whether inhibitory processing in depression is altered in a global motion processing task has not yet been established. Furthermore, existing studies have varied with regard to the medication status of their participants [11,12]. It is theorized that GABA is related to CSS as the primary inhibitory neurotransmitter in the brain [22,23], and GABA concentrations in the occipital cortex may be impacted by even non-GABAergic medications [24–26].”

One question I had reading the manuscript is how the authors tested the vision of the participants. Did they measure the spatial acuity for each eye separately? What instrument did they use? Did they measure the stereoacuity? This is an interesting point given that a recent study has shown that surround suppression is lower for monocular than binocular vision (Arranz-Paraíso et al. 2021). Thus, participants with an amblyopic eye could show lower suppression. Interestingly, ketamine that is used as an antidepressant in humans, is effective in treating adult amblyopia (Grieco et al., 2020). I wonder whether small eye deviations are more present in unmedicated depressed participants or if they present poorer stereoacuity. I think the authors should discuss this aspect in the paper.

Response: We thank the reviewer for this comment and for providing these references. We administered visual acuity binocularly, using the Freiburg visual acuity test, as now described in better detail in the methods. We also discuss the distinction of monocular versus binocular vision concerning center surround suppression in the discussion. 

Additions:

Page 7, “Visual acuity was assessed with the Freiburg visual acuity test [32].”

Page 20, “Research suggests that surround suppression is weaker for monocular than for binocular vision [51]. Further, connections between amblyopia (which would force perceivers to view stimuli monocularly) and depression have emerged in that both are treatable with ketamine. Future work would do well to explore CSS monocularly as well as binocularly in depressed participants, as well as to examine the effects various psychotropic medications not used by the participants in the present study, such as ketamine.”

Another recent study has shown differences in eye movements between MDD and controls (Takahashi et al, 2021). Is it possible that these differences in eye movements could explain these results for the unmedicated participants?

Response: We thank the reviewer for providing this reference. The study (Takashi et al., 2021) found that depressed participants had significantly shorter saccades than controls. While additional between-group differences were found for older participants (age>48), the average ages in our samples were below 40. While this opens an interesting line of thought, we ultimately do not believe our results are attributable to shorter saccades in the depressed group. Interestingly, a study by Hong et al. (2008) showed that altered open loop acceleration during smooth pursuit eye movement initiation may explain poor velocity discrimination in schizophrenia patients (Chen et al., 1999). While that result is not conclusive because of some key differences in stimulus conditions, it does open the possibility of motion performance being explained by eye movement differences in clinical samples. The present pattern of results seems unlikely to be explained by such effects, for two main reasons. First, because of the younger mean age of the depressed and healthy samples in the present study, we might not expect any difference in eye movements based on the Takashi et al., 2021 study. Secondly, if eye movement differences did occur between groups, we would expect it to affect motion perception performance in the baseline condition, as well as the Same and Opposite surround conditions. We now discuss this interesting literature in the discussion (page 20).

Addition:

Page 20, “Notably, other research has suggested that depressed individuals have significantly shorter saccades than controls in eye movement tests [50], which could be related to group differences in RDP direction of motion performance. However, if this were the case, we would expect to see differences in the Baseline condition, as well as the Same and Opposite conditions. Because this study only found differences in the Same condition, we do not believe saccade length is implicated in the depressed group’s performance.”

About the analysis, I would like to see in the main document the analysis including only the medicated participants compared with the healthy controls. One figure of accuracy and one of surround effects comparing controls and medicated participants would be very informative (e.g. Figures 4C and 5C).

Response: We thank the reviewer for their suggestion. We have updated figures 3, 4, and 5 to include a comparison of the medicated depressed participants and controls.

Why is the surround effect reduced with increasing coherence? Significant for facilitation but for suppression not significant if we take all coherences. I guess that if the authors fit regression lines to the data of Figures 5A and B, theses would be negative and probably significant. For suppression, the differences between 0.8% and 64% are clear, I would like the authors discuss this point.

Response: We thank the reviewer for raising this interesting point. The reduction in the surround effect is most likely due to the reduced difficulty of the task at higher coherencies (i.e., an accuracy ceiling effect). Participants’ responses become less variable, and less able to be impacted by the surround when the coherence is high. 

We have added notes to this effect in the caption of figure 5.

Additions: 

Figure 5 caption, page 15, “Suppression and facilitation effects likely decrease with greater coherence due to reduced difficulty of the task at higher coherence, which reduces variability of participant responses and the ability of the surround to influence responses.”

About the representation of the data, I think it is very informative to use beeswarm plot with boxplots to show whether there are outliers in the conditions. 

Response: We thank the reviewer for this suggestion. We also favor the idea of showing raw data whenever possible, and have updated the boxplot of BDI scores to 

include beeswarm plots. The plots in figures 3-5 are already quite busy with just the group means and different conditions being shown, and adding the individual data would make them difficulty to digest. We prefer not to add additional figures, as we already have a high number. We therefore choose to not use beeswarm format for the main results. However, to satisfy the reviewer’s curiosity, we have included boxplots with beeswarms for the baseline, same, and opposite conditions at the end of this letter.

Minor

-Regarding trials, although there is an image in Figure 1 describing them, I would like the authors to include a description of the trials in the main text. Why the presentation time is 250 msec? Have the authors tested other durations? The classical way to test motion surround suppression is measuring duration thresholds. Those are much lower than the duration used here.

Response: We thank the reviewer for his or interest in additional rationale for the task design. We have added a description of the task to the text, quoted below. In terms of the presentation time, the reviewer is correct that in studies like Tadin presentation time is the stimulus characteristic used to manipulate task difficulty. For motion integration tasks, the presentation time does not need to be short in order for the task to be difficult. We chose a brief presentation time (albeit longer than most of the presentation times in a grating direction discrimination task) and found that subjects still reached a floor at the lowest coherence level, and ceiling at the highest.

Addition: Page 8, “In each trial, a fixation cross was first displayed for 100 msec, followed by the RDP stimulus for 250 msec, followed by a blank screen until the participant responded (Figure 1). A brief stimulus presentation time (250 msec) was chosen to limit the role of eye movements during the task, while being sufficiently long to allow for high accuracy performance in the higher coherence conditions.”

-Page 9. How is the accuracy measured? Is it percentage of correct responses?

Response: We thank the reviewer for this question, to which the answer is yes. We have added a sentence to this effect to page 9 to confirm, quoted below. 

Addition: 

Page 9, “Participants’ accuracy in each task is represented by the proportion of correct responses out of the total number of trials.”

-Page 10. “…26 current (including 12 medicated), 20 remitted (including 9 medicated),…”

This is 26+20 =46 depressed participants. Thus, 46-(12+9)=25 unmedicated. In page 11 the authors say 27 unmedicated and also in the abstract.

Response: We thank the reviewer for catching this error. The correct number of medicated individuals in the remitted group is 7, rather than 9. We have fixed this oversight on page 8.

Revision: Page 7, “Nineteen participants were taking a psychotropic medication at the time of the study (n=12 currently depressed, n = 7 remitted depressed).”

-Page 11. For the ANOVAs. Did the authors test the normality of the data?

Response: We did test the normality of our data and found that the normality assumption did not hold. However, we moved forward with our planned analyses because statistical studies have found that Type I error in ANOVA tests is robust to non-normality (Blanca et al., 2017; Blanca et al., 2023). We have added a sentence explaining this to the results section, quoted below. 

Addition:

Page 11, “Shapiro-Wilk tests indicated that the data was not normal, but we moved forward with our planned ANOVAs because recent statistical studies have shown that Type I error in ANOVAs is robust to nonnormality [38,39].”

-Page 14. What is "higher facilitation"? Do you mean “lower Surround effect”?

Response: We thank the reviewer for this question. The facilitation effect is represented by the difference in accuracy between the Opposite condition and the Baseline condition. When the surrounding annulus moves in the opposite direction of the center, perception of the direction of motion is facilitated. This is not necessarily related to a lower surround effect. We have added a sentence clarifying this in the methods on p. 9, quoted below. 

Revision: 

Page 9, “CSF was calculated at each coherence level as the difference between Opposite accuracy and Baseline accuracy because improvements in accuracy (i.e., facilitation) tend to occur with a surrounding stimulus that moves in the opposite direction as the center motion [18]. CSS was calculated as the difference between Baseline accuracy and Same accuracy because decreases in accuracy (i.e., suppression) tend to occur when the surround and center move in the same direction [18].”

-Page 14-15. Are there differences between facilitation and suppression for the groups of participants tested?

Response: We thank the reviewer for this question. Our ANOVA analyses indicated that there was no main effect of group in facilitation or suppression when both medicated and unmedicated participants were included. When only unmedicated participants were included, there was no main effect of group for facilitation, but there was for suppression. Unmedicated depressed participants showed less suppression than controls. 

Relevant Text:

Page 14, “Including medicated and unmedicated participants… For facilitation … There was no significant main effect of group (F(1,72)=1.787, p=.186, η^2=0.024, 90% η^2 CI [0.00,0.108]). For suppression, there was no significant main effect of group (F(1,72)=1.272, p=.263, η^2=0.017, 90% η^2 CI [0.00,0.094])…”

Pages 14-15, “Including only unmedicated participants… For facilitation, … There was no significant main effect of group (F(1,53)=3.776, p=.057, η^2=0.067, 90% η^2 CI [0.00,0.191])… For suppression, there was a main effect of group (F(1,53)=5.420, p=.024, η^2=0.093, 90% η^2 CI [0.007,0.226]), with depressed individuals showing less suppression than controls (Fig 5-B).”

-Page 17. When the authors describe Golomb’s study, I think is better to talk about duration thresholds instead of accuracy.

Response: We appreciate the reviewer’s suggestion here, and he or she is correct that we mis-spoke. We meant to say, “better performance,” not “higher accuracy” when referencing the remitted depressed group in that study. We have revised our sentence on page 18, quoted below.

Revision:

Page 17, “Golomb et al. [11] found that unmedicated participants who were fully remitted from depression demonstrated significantly lower duration thresholds, i.e., improved performance compared with healthy controls when perceiving the direction of motion of a large grating stimulus. There was no significant difference with a small grating stimulus in the same participants.”

-Check the authors in the reference 19.

Response: We thank the reviewer for catching this typo, which has been corrected as shown below: 

Revision:

“Weiss Y, Adelson EH. Slow and Smooth: a Bayesian theory for the combination of local motion signals in human vision. 1998;1–41.”

References

Arranz-Paraíso, S., Read, J.C.A. & Serrano-Pedraza, I. (2021). Reduced surround suppression in monocular motion perception. Journal of Vision;21(1):10

Grieco SF, Qiao X, Zheng X, Liu Y, Chen L, Zhang H, Yu Z, Gavornik JP, Lai C, Gandhi SP, Holmes TC, Xu X. Subanesthetic Ketamine Reactivates Adult Cortical Plasticity to Restore Vision from Amblyopia. Curr Biol. 2020 Sep 21;30(18):3591-3603.e8. doi: 10.1016/j.cub.2020.07.008.

Pack CC, Hunter JN, Born RT. Contrast dependence of suppressive influences in cortical area MT of alert macaque. J Neurophysiol. 2005 Mar;93(3):1809-15. doi: 10.1152/jn.00629.2004. Epub 2004 Oct 13. PMID: 15483068.

Takahashi J, Hirano Y, Miura K, Morita K, Fujimoto M, Yamamori H, Yasuda Y, Kudo N, Shishido E, Okazaki K, Shiino T, Nakao T, Kasai K, Hashimoto R and Onitsuka T (2021) Eye Movement Abnormalities in Major Depressive Disorder. Front. Psychiatry 12:673443

Reviewer #2: 

In the manuscript entitle ‘Reduced visual context effects in global motion processing in depression’, Murray & Norton study visual motion perception in patients with depression. They find altered motion processing in depression, but only on those patients without medication. This is a nice addition to the literature on visual abnormalities in depression. The manuscript is well-written, methods are technically sound, and results are clearly presented. I have only a few minor comments, presented below.

1) How the exclusion criteria were checked? Only subjective report or other means, please clarify.

Response: We thank the reviewer for this question. Exclusion criteria were verified through phone call screenings and in-person interviews and questionnaires, including the SCID-IV. Visual acuity was assessed with the Freiburg visual acuity test. 

Addition: Page 7, “Exclusion criteria were verified through phone call screenings and in-person interviews and questionnaires, including the Structured Clinical Interview for DSM-4 (SCID) [31]. Visual acuity was assessed with the Freiburg visual acuity test [32].”

2) Why the analysis were repeated separately with medicated and unmedicated? The other option would have been medication as an additional between-subjects factor. Was there any reason for separate analyses?

Response: We thank the reviewers for this question and helpful suggestion. Our main reason for running separate analyses was that no healthy participants in our sample were taking psychotropic medications. While there may be some existing methods to get around the issue of an empty cell in an ANOVA, we ultimately decided to move forward with two separate comparisons for the sake of interpretability. For example, an interaction between group, medication, coherence and surround condition would be difficult to interpret, especially since there is no variability in medication status in the control group. In addition, our study was underpowered to address 4 factors at once in an analysis. 

3) The authors could consider linear mixed effects model instead of repeated measures ANOVA. Some of the excluded participants could be possible added to the analysis, since the mixed model can model data with missing values. The linear mixed effects model also has larger statistical power than rmANOVA. And perhaps the total number of statistical tests in the manuscript (and supplement) could also be reduced with a little bit more complex statistical model.

Response: We thank the reviewer for this suggestion. In our dataset, if a participant was missing data for one coherence level in a condition, they were missing data for all coherence levels in that condition. Similarly, if a participant was missing data for the Same condition, they were also missing data for the Opposite condition, and vice versa. For this reason, a linear mixed effects model would not provide an advantage in terms of handling missingness. We also encounter a similar problem to the one mentioned above regarding the 4-way ANOVA. Because there are no medicated healthy controls, the results are harder to interpret. For example, a main effect of medication status in this analysis is comparing medicated depressed participants to both unmedicated depressed participants and healthy controls. Therefore, for the sake of simplicity and interpretability, we decided to move forward with our existing ANOVAs.

4) Quite a few ANOVAs were conducted. Did the p-values remain significant after correction for multiple comparisons? For example, FDR-correction across all the reported p-values would be simple to calculate.

Response: We thank the reviewer for this suggestion. We have conducted the Benjamin-Hochberg procedure across our 37 reported p-values (including the tests of accuracy between remitted and currently depressed, main effects and interactions within ANOVAs, and post hoc t-tests) with a false discovery rate of .10. We found that all significant p-values remained significant with the FDR correction.

 Our largest p-value that is less than its corresponding q-value is 0.024 < 0.035. This is also our largest significant un-corrected p-value at the alpha = .05 level, thus all smaller p-values remain significant. 

Addition: Page 9, “We conducted a False Discovery Rate (FDR) correction using the Benjamin-Hochberg procedure [37] with an FDR of 0.10. All p-values reported below remained significant with the FDR correction.”

5) In discussion, the SSN model is presented. What then would be the cause of abnormal motion perception in depression, altered neuronal gain or network dynamics or something else? Could the authors speculate a bit more based on the SSN model?

Response: We thank the reviewer for their interest and suggestion in this interesting model. We have added a few additional sentences on the SSN on pages 19-20.

Addition: Pages 19-20, “Whether the SSN can account for differences in CSS between depressed and healthy individuals remains an open question. It is possible that subtle changes in network dynamics of the supralinear or sublinear portions of neuronal response to strong and weak stimuli can explain some of the results thus far. For example, a higher breaking point before the system switches from supralinear to sublinear behavior, or weaker gain for neuronal response as external input strength increases, could explain the reduced CSS effects shown in this and prior studies. Given that one study found increased CSS effects in some conditions, and trends towards reduced ones in others [10], this will be an interesting topic to test and explore in different aspects of the visual system. Similar work investigating the effects of other neurotransmitters on polysynaptic summation might contribute to a better understanding of the differences in inhibitory processes between depressed and healthy brains.”

References

Blanca Mena, M. J., Alarcón Postigo, R., Arnau Gras, J., Bono Cabré, R., & Bendayan, R. (2017). Non-normal data: Is ANOVA still a valid option?. Psicothema, 2017, vol. 29, num. 4, p. 552-557.

Blanca Mena, M. J., Arnau Gras, J., García de Castro, F. J., Alarcón Postigo, R., & Bono Cabré, R. (2023). Non-normal data in repeated measures ANOVA: impact on type I error and power. Psicothema.

---

## [Decision Letter · Decision Letter 1]

10 Aug 2023

PONE-D-23-03585R1Reduced visual context effects in global motion processing in depressionPLOS ONE

Dear Dr. Norton,

Thank you for submitting your manuscript to PLOS ONE. After careful consideration, we feel that it has merit but does not fully meet PLOS ONE’s publication criteria as it currently stands. Therefore, we invite you to submit a revised version of the manuscript that addresses the points raised during the review process.

We look forward to receiving your revised manuscript.

Kind regards,

Giulia Prete

Academic Editor

PLOS ONE

Journal Requirements:

**Additional Editor Comments:**

In particular, both previous Reviewers agreed to consider the revised version of the manuscript. Reviewer 2 suggests to accept the manuscript in its present form, Reviewer 1 recognizes the improvement due to the revisions, but s/he still suggest a further point which, in my opinion, can help in further improve the clarity of the study. Thus, I would like to ask you to reply to this latter point and to re-submit the new version, which I will consider by myself if it will be the case. 

Reviewers' comments:

Reviewer's Responses to Questions

**Comments to the Author**

1. If the authors have adequately addressed your comments raised in a previous round of review and you feel that this manuscript is now acceptable for publication, you may indicate that here to bypass the “Comments to the Author” section, enter your conflict of interest statement in the “Confidential to Editor” section, and submit your "Accept" recommendation.

Reviewer #1: All comments have been addressed

Reviewer #2: All comments have been addressed

2. Is the manuscript technically sound, and do the data support the conclusions?

Reviewer #1: Yes

Reviewer #2: Yes

3. Has the statistical analysis been performed appropriately and rigorously? 

Reviewer #1: Yes

Reviewer #2: Yes

4. Have the authors made all data underlying the findings in their manuscript fully available?

Reviewer #1: Yes

Reviewer #2: Yes

5. Is the manuscript presented in an intelligible fashion and written in standard English?

Reviewer #1: Yes

Reviewer #2: Yes

6. Review Comments to the Author

Reviewer #1: I think the authors have done a good work with the review and the paper is much better now, but I still have a few comments.

Regarding the local-global motion I think it is better explained now in the paper. Regarding the discussion about the link between GABA and CSS (lines 412) there are few studies that are worth to be mentioned too, for example, Schallmo et al. (2018), that combined psychophysics and magnetic resonance spectroscopy and found that suppression in humans is not primarily driven by GABAergic inhibition; Liu & Pack (2014), showed that manipulations of GABA levels in MT did not have effect on surround suppression; and Read et al. (2015) that found that acute alcohol intoxication had no effect on surround suppression (low alcohol concentrations enhance the inhibition of the GABAergic system).

Interestingly, Schallmo et al. (2018) proposed a single computational principle (divisive normalization) that could account both facilitation and suppression. According to them, “there is no need to invoke separate neural mechanisms”. I wonder how a single mechanism for facilitation and suppression could explain the findings of this current paper where facilitation and suppression are affected differentially for unmedicated/medicated participants. Is it possible that your findings provide evidence of distinct neural mechanisms for facilitation and suppression? Could you discuss this?

References

Schallmo MP, Kale AM, Millin R, Flevaris AV, Brkanac Z, Edden RA, Bernier RA, Murray SO. (2018). Suppression and facilitation of human neural responses. Elife. doi: 10.7554/eLife.30334.

Liu L, Pack C. (2014) Bidirectional manipulation of GABAergic inhibition in MT: A comparison of neuronal and psychophysical performance. Journal of Vision. 2014; 14(10):13-. https://doi.org/10.1167/14.10.13

Read JCA, Georgiou R, Brash C, Yazdani P, Whittaker R, Trevelyan AJ, et al. (2015). Moderate acute alcohol intoxication has minimal effect on surround suppression measured with a motion direction discrimination task. Journal of vision. 15(1):15.1.5. https://doi.org/10.1167/15.1.5

Reviewer #2: (No Response)

7. PLOS authors have the option to publish the peer review history of their article (what does this mean?). If published, this will include your full peer review and any attached files.

Reviewer #1: No

Reviewer #2: No

---

## [Author Response · Author response to Decision Letter 1]

28 Aug 2023

August 28, 2023

To the editorial team of Plos One,

We are pleased to submit our second round of revisions to this human subjects research article entitled, “Reduced visual context effects in global motion processing in depression” for consideration of publication in PLOS ONE.

We sincerely thank you and the reviewers for the thoughtful comments on our manuscript, which we have addressed to the best of our ability.

All comments from Reviewer 1 are listed below, and Reviewer 2 did not have any additional comments. For each, our response is shown indented and in italics, and the corresponding changes to the manuscript are indicated. We have also included our updated reference list at the end of this letter.

Sincerely,

Daniel J. Norton, Ph.D.

Assistant Professor of Psychology | Gordon College 

danieljnorton@gmail.com | 617-855-4431

Grace E. Murray, M.A.

PhD Student in Clinical Psychology | Boston University

gemurray@bu.edu

COMMENTS TO THE AUTHOR

Reviewer #1: 

I think the authors have done a good work with the review and the paper is much better now, but I still have a few comments.

Regarding the local-global motion I think it is better explained now in the paper. Regarding the discussion about the link between GABA and CSS (lines 412) there are few studies that are worth to be mentioned too, for example, Schallmo et al. (2018), that combined psychophysics and magnetic resonance spectroscopy and found that suppression in humans is not primarily driven by GABAergic inhibition; Liu & Pack (2014), showed that manipulations of GABA levels in MT did not have effect on surround suppression; and Read et al. (2015) that found that acute alcohol intoxication had no effect on surround suppression (low alcohol concentrations enhance the inhibition of the GABAergic system).

Interestingly, Schallmo et al. (2018) proposed a single computational principle (divisive normalization) that could account both facilitation and suppression. According to them, “there is no need to invoke separate neural mechanisms”. I wonder how a single mechanism for facilitation and suppression could explain the findings of this current paper where facilitation and suppression are affected differentially for unmedicated/medicated participants. Is it possible that your findings provide evidence of distinct neural mechanisms for facilitation and suppression? Could you discuss this?

Response: We thank the reviewer for these interesting points, and for providing these 

references. We have added citations of Read et al., 2015 and Liu & Pack, 2014 to page 19, enhancing our argument. We have also added a brief discussion of Schallmo et al’s 2018 paper and its implications regarding the current study.

Additions & Revisions:

Page 19, “Another study found that alcohol administration did not lead to an increased suppression index, which would be expected given alcohol’s potentiation of GABA-ergic transmission [49].”

Page 19, “First, local blockage of GABA inputs does not disrupt motion suppression [44, 50].”

Page 20, “Schallmo et al. [52] have proposed an additional model of suppression and facilitation that does not rely principally on GABAergic inhibition. Their theory makes use of divisive normalization, the idea that neuronal output depends not only on the input, but also on the activity of surrounding neurons [52]. They propose a model of neuronal response based on the ratio between an excitatory drive, defined by the strength of the input (i.e., contrast level), and a suppressive drive [52]. This model can explain both CSF and CSS, eliminating the need for separate neural mechanisms for facilitatory and suppressive behavior (see [52] for a full explanation). However, this model does not explain the results of the present study, which indicate differences in CSS, but not CSF, in unmedicated depressed and healthy adults. Future research is needed to determine if the results of the present study are evidence of distinct neural mechanisms of CSS and CSF, or if alterations in the relative strength of suppressive or faciliatory contextual inputs in depression can be explained through the divisive normalization model. “

References

Schallmo MP, Kale AM, Millin R, Flevaris AV, Brkanac Z, Edden RA, Bernier RA, Murray SO. (2018). Suppression and facilitation of human neural responses. Elife. doi: 10.7554/eLife.30334.

Liu L, Pack C. (2014) Bidirectional manipulation of GABAergic inhibition in MT: A comparison of neuronal and psychophysical performance. Journal of Vision. 2014; 14(10):13-. https://doi.org/10.1167/14.10.13

Read JCA, Georgiou R, Brash C, Yazdani P, Whittaker R, Trevelyan AJ, et al. (2015). Moderate acute alcohol intoxication has minimal effect on surround suppression measured with a motion direction discrimination task. Journal of vision. 15(1):15.1.5. https://doi.org/10.1167/15.1.5

References

[1] Hasin DS, Grant BF. The National Epidemiologic Survey on Alcohol and Related Conditions (NESARC) Waves 1 and 2: review and summary of findings. Soc Psychiatry Psychiatr Epidemiol 2015;50:1609–40. https://doi.org/10.1007/s00127-015-1088-0.

[2] Grahek I, Shenhav A, Musslick S, Krebs RM, Koster EHW. Motivation and cognitive control in depression. Neurosci Biobehav Rev 2019;102:371–81. https://doi.org/10.1016/j.neubiorev.2019.04.011.

[3] Kircanski K, Joormann J, Gotlib IH. Cognitive aspects of depression. Wiley Interdiscip Rev Cogn Sci 2012;3:301–13. https://doi.org/10.1002/wcs.1177.

[4] Akil H, Gordon J, Hen R, Javitch J, Mayberg H, McEwen B, et al. Treatment resistant depression: A multi-scale, systems biology approach. Neurosci Biobehav Rev 2018;84:272–88. https://doi.org/10.1016/j.neubiorev.2017.08.019.

[5] Treadway MT, Pizzagalli DA. Imaging the pathophysiology of major depressive disorder-from localist models to circuit-based analysis. Biol Mood Anxiety Disord 2014.

[6] Fitzgerald PJ. Gray colored glasses: Is major depression partially a sensory perceptual disorder? J Affect Disord 2013;151:418–22. https://doi.org/10.1016/j.jad.2013.06.045.

[7] Hubel DH, Wiesel TN. Receptive fields of single neurones in the cat’s striate cortex. J Physiol 1959;48:574–91.

[8] Wallisch P, Kumbhani RD. Can major depression improve the perception of visual motion? Journal of Neuroscience 2009;29:14381–2. https://doi.org/10.1523/JNEUROSCI.4560-09.2009.

[9] Salmela V, Socada L, Söderholm J, Heikkilä R, Lahti J, Ekelund J, et al. Reduced visual contrast suppression during major depressive episodes. Journal of Psychiatry and Neuroscience 2021;46:E222–31. https://doi.org/10.1503/jpn.200091.

[10] Norton DJ, McBain RK, Pizzagalli DA, Cronin-Golomb A, Chen Y. Dysregulation of visual motion inhibition in major depression. Psychiatry Res 2016;240:214–21. https://doi.org/10.1016/j.psychres.2016.04.028.

[11] Golomb JD, McDavitt JRB, Ruf BM, Chen JI, Saricicek A, Maloney KH, et al. Enhanced visual motion perception in major depressive disorder. Journal of Neuroscience 2009;29:9072–7. https://doi.org/10.1523/JNEUROSCI.1003-09.2009.

[12] Song XM, Hu XW, Li Z, Gao Y, Ju X, Liu DY, et al. Reduction of higher-order occipital GABA and impaired visual perception in acute major depressive disorder. Mol Psychiatry 2021;26:6747–55. https://doi.org/10.1038/s41380-021-01090-5.

[13] Allman J, Miezin F, Mcguinness E. Stimulus Specific responses from Beyond the Classical Receptive Field: Neurophysiological Mechanisms for Local-Global Comparisons in Visual Neurons. Annu Rev Neurosci 1985:407–30.

[14] Born RT. Center-Surround Interactions in the Middle Temporal Visual Area of the Owl Monkey 2000:2658–69.

[15] Evers K, Peters J, Senden M. Cortical Synchrony as a Mechanism of Collinear Facilitation and Suppression in Early Visual Cortex. Front Syst Neurosci 2021;15:1–12. https://doi.org/10.3389/fnsys.2021.670702.

[16] Tadin D, Lappin JS, Gilroy LA, Blake R. Perceptual consequences of centre-surround antagonism in visual motion processing. Nature 2003;424:312–5. https://doi.org/10.1038/nature01812.

[17] Ichida JM, Schwabe L, Bressloff PC, Angelucci A. Response facilitation from the “suppressive” receptive field surround of macaque V1 neurons. J Neurophysiol 2007;98:2168–81. https://doi.org/10.1152/jn.00298.2007.

[18] Born RT, Bradley DC. Structure and function of visual area MT. Annu Rev Neurosci 2005;28:157–89. https://doi.org/10.1146/annurev.neuro.26.041002.131052.

[19] Williams DW&, Sekuler R. Coherent global motion percepts from stochastic local motions. ACM SIGGRAPH Computer Graphics, 1984, p. 24.

[20] Chen Y, Nakayama K, Levy D, Matthysse S, Holzman P. Processing of global, but not local, motion direction is deficient in schizophrenia. Schizophr Res 2003;61:215–27. https://doi.org/10.1016/S0920-9964(02)00222-0.

[21] Adelson E, Movshon JA. Phenomenal coherence of moving visual patterns. Nature 1982;300:523–5.

[22] Cook E, Hammett ST, Larsson J. GABA predicts visual intelligence. Neurosci Lett 2016;632:50–4. https://doi.org/10.1016/j.neulet.2016.07.053.

[23] Somogyi P, Tamas G, Lujan R, Buhl EH. Salient features of synaptic organisation in the cerebral cortex 1. Brain Res Rev 1998;26:113–35.

[24] Bhagwagar Z, Marzena Wylezinska Mrcp, Taylor M, Peter Jezzard M, Matthews PM, Philip Cowen FJ. Increased Brain GABA Concentrations Following Acute Administration of a Selective Serotonin Reuptake Inhibitor. Am J Psychiatry 2004;161:368–70.

[25] Popov N, Matthies H. Some effects of monoamine oxidase inhibitors on the metabolism of 7-aminobutyric acid in rat brain. Journal of Neurwhemistry 1969;16:899–907.

[26] Patel GJ, Schatz RP, Constantinides SM, Lal H. Effect of desipramine and pargyline on brain gamma-aminobutryic acid. Biochem Pharmacol 1975;24:57–60.

[27] Newsome WT, Pare EB. A Selective Impairment of Motion Perception Following Lesions of the Middle Temporal Visual Area (MT). The Journal of Neuroscience 1988;8:2201–11.

[28] Born RT, Tootell RBH. Segregation of global and local motion processing in primate middle temporal visual area. Nature 1992;357:497–9.

[29] Keane BP, Paterno D, Crespo LP, Kastner S, Silverstein SM. Smaller visual arrays are harder to integrate in schizophrenia: Evidence for impaired lateral connections in early vision. Psychiatry Res 2019;282. https://doi.org/10.1016/j.psychres.2019.112636.

[30] Chen Y, Norton D, Ongur D. Altered Center-Surround Motion Inhibition in Schizophrenia. Biol Psychiatry 2008;64:74–7. https://doi.org/10.1016/j.biopsych.2007.11.017.

[31] First MB, Spitzer RL, Gibbon M, Williams JB. Structured clinical interview for DSM-IV-TR axis I disorders, research version, patient edition. 2002.

[32] Bach M. FrACT-Landolt-Vision. Optometry and Visual Science 1996;73:49–53.

[33] Wilkinson G s. WRAT-3: Wide range achievement test administration manual. Wilmington, DE: 1993.

[34] Beck AT, Steer RA, Brown GK. Beck depression inventory (BDI-II). London, UK: Pearson; 1996.

[35] Rush AJ, Trivedi MH, Ibrahim HM, Carmody TJ, Arnow B, Klein DN, et al. The 16-Item Quick Inventory of Depressive Symptomatology (QIDS), Clinician Rating (QIDS-C), and Self-Report (QIDS-SR): A Psychometric Evaluation in Patients with Chronic Major Depression. Biol Psychiatry 2003;54:573–83. https://doi.org/10.1016/S0006-3223(03)01866-8.

[36] Beck AI, Steer RA, Carbin MC. Psychometric propertis of the Beck Depression Inventory: Twenty-five years of evaluation. Clin Psychol Rev 1988;8:77–100.

[37] Thissen D, Steinberg L, Kuang D. Quick and Easy Implementation of the Benjamini-Hochberg Procedure for Controlling the False Positive Rate in Multiple Comparisons. n.d.

[38] Blanca MJ, Arnau J, García-Castro FJ, Alarcón R, Bono R. Non-normal Data in Repeated Measures ANOVA: Impact on Type I Error and Power. Psicothema 2023;35:21–9. https://doi.org/10.7334/psicothema2022.292.

[39] Blanca MJ, Alarcón R, Arnau J, Bono R, Bendayan R. Non-normal data: Is ANOVA still a valid option? Psicothema 2017;29:552–7. https://doi.org/10.7334/psicothema2016.383.

[40] Norton D, McBain R, Holt DJ, Ongur D, Chen Y. Association of Impaired Facial Affect Recognition with Basic Facial and Visual Processing Deficits in Schizophrenia. Biol Psychiatry 2009;65:1094–8. https://doi.org/10.1016/j.biopsych.2009.01.026.

[41] Wesner MF, Tan J. Contrast sensitivity in seasonal and nonseasonal depression. J Affect Disord 2006;95:19–28. https://doi.org/10.1016/j.jad.2006.03.028.

[42] Berezovskii VK, Born RT. Specificity of Projections from Wide-Field and Local Motion-Processing Regions within the Middle Temporal Visual Area of the Owl Monkey. The Journal of Neuroscience 2000;20:1157–69.

[43] Enroth-Cugell C, Robson JG. The contrast sensitivity of retinal ganglion cells of the cat. J Physiol 1966;187:517–52.

[44] Furlan M, Smith AT. Global motion processing in human visual cortical areas V2 and V3. Journal of Neuroscience 2016;36:7314–24. https://doi.org/10.1523/JNEUROSCI.0025-16.2016.

[45] Veselinović T, Schorn H, Vernaleken IB, Hiemke C, Zernig G, Gur R, et al. Effects of antipsychotic treatment on cognition in healthy subjects. Journal of Psychopharmacology 2013;27:374–85. https://doi.org/10.1177/0269881112466183.

[46] Chapman LJ, Chapman JP. Problems in the measurement of cognitive deficit. Psychol Bull 1973;79:380–5.

[47] Truong V, Cheng PZ, Lee HC, Lane TJ, Hsu TY, Duncan NW. Occipital gamma-aminobutyric acid and glutamate-glutamine alterations in major depressive disorder: An mrs study and meta-analysis. Psychiatry Res Neuroimaging 2021;308. https://doi.org/10.1016/j.pscychresns.2020.111238.

[48] Liu LD, Miller KD, Pack CC. A unifying motif for spatial and directional surround suppression. Journal of Neuroscience 2018;38:989–99. https://doi.org/10.1523/JNEUROSCI.2386-17.2017.

[49] Read, J. C., Georgiou, R., Brash, C., Yazdani, P., Whittaker, R., Trevelyan, A. J., & Serrano-Pedraza, I. (2015). Moderate acute alcohol intoxication has minimal effect on surround suppression measured with a motion direction discrimination task. Journal of vision 2015;15. https://doi.org/10.1167/15.1.5

[50] Liu L, Pack C. Bidirectional manipulation of GABAergic inhibition in MT: A comparison of neuronal and psychophysical performance. Journal of Vision 2014;14(10).

[51] Rubin DB, Van Hooser SD, Miller KD. The Stabilized Supralinear Network: A Unifying Circuit Motif Underlying Multi-Input Integration in Sensory Cortex. Neuron 2015;85:402–17. https://doi.org/10.1016/j.neuron.2014.12.026.

[52] Schallmo MP, Kale AM, Millin R, Flevaris AV, Brkanac Z, Edden RA, Bernier RA, Murray SO. Suppression and facilitation of human neural responses. Elife; 2018;7:e30334.

[53] Takahashi J, Hirano Y, Miura K, Morita K, Fujimoto M, Yamamori H, et al. Eye Movement Abnormalities in Major Depressive Disorder. Front Psychiatry 2021;12. https://doi.org/10.3389/fpsyt.2021.673443.

[54] Arranz-Paraíso S, Read JCA, Serrano-Pedraza I. Reduced surround suppression in monocular motion perception. J Vis 2021;21:1–16. https://doi.org/10.1167/jov.21.1.10.

---

## [Editor Report · Decision Letter 2]

31 Aug 2023

Reduced visual context effects in global motion processing in depression

PONE-D-23-03585R2

Dear Dr. Norton,

We’re pleased to inform you that your manuscript has been judged scientifically suitable for publication and will be formally accepted for publication once it meets all outstanding technical requirements.

Kind regards,

Giulia Prete

Academic Editor

PLOS ONE

Additional Editor Comments (optional):

I would like to thank you again for having the patience to reach this final decision. I am very satisfied with the final result and with your work and I am grateful for your choice to publish with us. Good luck for your work!

---

## [Editor Report · Acceptance letter]

5 Sep 2023

PONE-D-23-03585R2 

Reduced visual context effects in global motion processing in depression 

Dear Dr. Norton:

I'm pleased to inform you that your manuscript has been deemed suitable for publication in PLOS ONE. Congratulations! Your manuscript is now with our production department. 

Kind regards, 

on behalf of

Dr. Giulia Prete 

Academic Editor

PLOS ONE